# Metal Imbalance in Neurodegenerative Diseases with a Specific Concern to the Brain of Multiple Sclerosis Patients

**DOI:** 10.3390/ijms21239105

**Published:** 2020-11-30

**Authors:** Jean-Philippe Dales, Sophie Desplat-Jégo

**Affiliations:** 1Institute of Neurophysiopathology, CNRS, INP, Aix-Marseille University, 13005 Marseille, France; sophie.DESPLAT@ap-hm.fr; 2Assistance Publique-Hôpitaux de Marseille, Hôpital Nord, Pavillon Etoile, Pôle de Biologie, Service d’anatomie-pathologie, CEDEX 20, 13915 Marseille, France; 3Assistance Publique-Hôpitaux de Marseille, Hôpital de la Conception, Pôle de Biologie, Service d’Immunologie, 13005 Marseille, France

**Keywords:** multiple sclerosis, central nervous system, chemical element, metal, oxidative stress

## Abstract

There is increasing evidence that deregulation of metals contributes to a vast range of neurodegenerative diseases including multiple sclerosis (MS). MS is a chronic inflammatory disease of the central nervous system (CNS) manifesting disability and neurological symptoms. The precise origin of MS is unknown, but the disease is characterized by focal inflammatory lesions in the CNS associated with an autoimmune reaction against myelin. The treatment of this disease has mainly been based on the prescription of immunosuppressive and immune-modulating agents. However, the rate of progressive disability and early mortality is still worrisome. Metals may represent new diagnostic and predictive markers of severity and disability as well as innovative candidate drug targets for future therapies. In this review, we describe the recent advances in our understanding on the role of metals in brain disorders of neurodegenerative diseases and MS patients.

## 1. Introduction

Metals are vital for several biochemical processes and are required for normal central nervous system (CNS) functioning [1,2,3]. They are natural constituents of the earth’s soil and are disseminated into the biosphere through human activities [4]. Some metals are of bio-importance to health and their daily medicinal and dietary allowances have been recommended. They must be obtained from the environment and appropriately bound or compartmentalized within cells and tissues for use in biochemical pathways. They are essential cofactors for enzymes and structural elements notably implicated in the synthesis, stability, and maintenance of myelin [5,6,7]. Dietary intakes of these metals have to be maintained at regulatory limits, as excesses can result in toxicity. Other metals considered as xenobiotics have no known bio-importance in human physiology and their consumption even at very low concentrations can be detrimental. The leading mechanisms by which metals may induce autoimmune process are not well known, but they seem to imply a molecular mimicry, which occurs when similarities between foreign-derived and self-peptides favor an activation of autoreactive immune cells in susceptible individuals. It has been postulated that some xenobiotic metals may generate neoantigens by altering self-proteins both directly and indirectly or involve the exposure of parts of self-proteins not previously presented to the immune system, i.e., cryptic epitopes, which are therefore identified as non-self [8,9]. Thus, these metals can initiate an inappropriate T cell response in several ways, such as post-translational conformational modification of a bound peptide or altered antigen presentation to the Human leukocyte antigen (HLA) molecules.

There is now increasing evidence that deregulation of metals, such as iron, copper, and zinc homeostasis, contributes to a vast range of neurodegenerative diseases [10,11,12,13,14]. Multiple sclerosis (MS) is a chronic immune-mediated disease of the CNS in which an inflammatory process results in myelin sheath destruction, gliosis, and neurodegeneration. Although the origin of MS is still unknown, genetic predisposition and environmental toxicity may activate the immune system against CNS cells. Some metals such as mercury from dental amalgam have been discussed in the past as controversial etiologic factor in MS [15,16]. In addition, patients with relapsing remitting and secondary progressive MS have been shown to excrete large amounts of aluminum in their urine [17,18]. In MS patients, deregulation of metals in the serum, the cerebrospinal fluid (CSF), and the brain have been reported, however it is unclear whether it represents an initial or a subsequent event of the disease [19,20,21].

The aim of this work is to provide a summary of the current literature around the whole metal abnormalities in neurodegenerative disease patients and a meaningful review of the studies of these abnormalities in the blood and CNS in MS patients. Previous reviews in this area have largely focused on the potential role for isolated or a few metals in the pathogenesis, but to our knowledge, no study has encompassed a comprehensive analysis of the entirety of metals. It appears useful to provide a global view of the known metal imbalance, because metals are in a fine tuning equilibrium with each other in human tissues and deep understanding of the complexity of this imbalance may implicate to explore metals in their wholeness. Otherwise, literature studies about magnetic resonance imaging (MRI) techniques, which have been previously well reviewed will be out of the scope of this work. In this context, we will also develop perspectives on the future of “metals and MS” based around the bio-imaging of metals to further understand the mechanisms of the disease and possibly to identify potential targets for innovative treatments.

## 2. Essential Metals in the Normal Brain

### 2.1. Metal Homeostasis and Global Content

Essential metals are those chemical elements that are required for the normal functioning of biological tissues such as brain parenchyma. These metals in biological systems may be broadly divided into two groups: essential alkali and alkaline-earth metals such as sodium, potassium, magnesium, and calcium; essential transition metals such as copper, manganese, iron, zinc, and less abundant chromium, cobalt, molybdenum, or nickel. These metals are incorporated in a whole host of physiological processes such as electron and oxygen transport, protein modification, neurotransmitter synthesis, redox reactions, cell adhesion, protein, and carbohydrate metabolism and immune response.

Transition metals are known to be present in protein active sites as metabolic cofactors for structural and catalytic functions, and they are increasingly also recognized for a second messenger role in cell signaling [22,23]. In the CNS, iron, copper, and zinc metals act as essential cofactors in metalloproteins with unique importance in myelin synthesis, maintenance of oligodendrocytes, and neurotransmitter synthesis [1,6,7,13,22,23,24,25,26,27]. For example, the functions of iron include oxygen homeostasis, energy metabolism, DNA synthesis and repair, myelination of the white matter as well as synthesis, uptake, and degradation of neurotransmitters. Copper’s functions include synthesis of catecholamines, activation of neuropeptides and hormones, antioxidant defense, connective tissue production, synaptic transmission, and immune function [28,29,30]. Manganese appears mainly bound to the astrocyte-specific enzyme glutamine synthetase, and it can have profound effects on microglia and astrocytes by amplifying the activation of a proinflammatory response [31,32,33]. Zinc is a structural and catalytic component contributing to the efficient performance of over 2000 transcription factors and more than 300 enzymes that are essential for antioxidant response as well as acquired and innate immune responses [34,35,36].

Metals can exist in many different forms within the cells including as free ions. These ions are coordinately incorporated in biomolecules such as proteins or in labile association with low-molecular-weight species such as amino acids or glutathione from which the metal can be released by changes in the cellular environment [37]. They show transient changes in concentration occurring as a result of exchange between these metal-ion-binding species and labile metal ion pools within the cells [38]. A highly conserved group of proteins strictly controls the homeostasis of the essential metal ions by controlling their intake, intracellular distribution, storage, and export [22,39,40]. Their transport into the brain is also strictly regulated by the brain barrier system, i.e., the blood–brain and blood–cerebrospinal fluid (CSF) barriers [41]. For example, iron homeostasis is regulated by a complex network involving multiple transporter proteins including transferrin receptor 1, divalent metal transporter 1 (DMT1), lactoferrin, melanotransferrin, and ferroportin [42]. The transport of zinc into the brain occurs via its binding with L-histidine in target sites that regulates its uptake across the brain barrier system [43]. Zinc homeostasis in the brain is also tightly regulated primarily via three families of proteins: (1) the metallothioneins, which are involved in the regulation and maintenance of intracellular zinc content [44,45]; (2) the zinc- and iron-like regulatory proteins, which are responsible for zinc uptake from extracellular fluids into both neurons and glia [46]; (3) the zinc transporters, which are associated with cellular zinc efflux [47]. Interestingly, many of these proteins regulate other metal ions transport. In the brain, zinc is also present in a free ionic form (Zn^2+^) and enriched within synaptic vesicles at glutamatergic nerve terminals from where it is synaptically released during neuronal activity [48,49].

Some studies have previously reported the levels at which metals are present within the brain parenchyma with various concentrations depending on the analytical methods that were used [50,51]. The literature values of concentration of various elements found by different research groups have been reviewed by Grochowski et al. [50]. In these studies, it appeared that the spectroscopic methods (e.g., paragraph 5) were the most often used for quantitative and qualitative analysis of metals in human brain samples probably due to the low limits of quantification, the ability to determine most elements of the periodic table, and the relatively short analysis time [50]. Calcium and manganese appeared to be the most abundant elements with reported concentrations as high as 50–150 µg/g per dry weight in the brain of healthy adults followed by iron (20 to 100 µg/g) and zinc (about 10 µg/g) [50]. Copper and magnesium levels were found lower than 10 µg/g. Increased iron and copper metal contents have also been documented in aging brain, where iron accumulates in microglia and astrocytes [52,53,54]. Such metal accumulations may be attributed to an increase in age-related oxidative stress as well as a progressive alteration of protein pathways that regulate their trafficking.

### 2.2. Regional Heterogeneity in Brain Parenchyma

Concentrations of metals in the brain are highly compartmentalized [51,55,56]. Accordingly, iron is present primarily in oligodendrocytes and myelin, where it is stored predominantly in its redox-inactive ferric (Fe^3+^) form within ferritin [15,53]. The gray matter structures generally have higher concentrations of iron, copper, and potassium and lower concentrations of zinc than white matter, because myelin is rich in zinc, which stabilizes its structure [57,58,59,60]. The brain regions associated with motor functions contain two to three times more iron than non-motor-related regions [12]. Substantia nigra, globus pallidus, putamen, caudate nucleus, red nucleus, dentate nucleus, and locus coeruleus contain the highest concentrations of iron. Copper levels have been reported higher in the cortex and the white matter of cerebellum than in cerebrum ones [57]. They are high in the substantia nigra, locus coeruleus, dentate nucleus, and cerebellum [57,61,62,63]. In addition, while most iron- and copper-rich brain structures are low in zinc, the hippocampus and the amygdala have been shown to be rich in zinc metal, because these regions have numerous zincergic neurons [58,60,64,65]. Furthermore, high levels of manganese have been reported in the corpus pineale [57,66].

## 3. Metals in Neurodegenerative Diseases and Neuroinflammation

### 3.1. Essential Metal Dyshomeostasis: A Cause or a Consequence?

There is considerable evidence to suggest that homeostasis of metals ions play critical roles in disorders of the CNS including Alzheimer disease (AD), Parkinson’s disease (PD), amyotrophic lateral sclerosis (ALS), multiple system atrophy, or prion diseases [67,68,69,70]. Alterations in the handling or increased accumulation of essential metals have been well reported to exert neurotoxicity [13]. Chronic exposure to essential metal such as manganese has been shown to result in Parkinson-like syndrome referred to as “manganism” [71]. The neurotoxicity induced by the overexposure of this metal includes disruption of mitochondrial function, disruption of neurotransmitter metabolism, alteration of iron homeostasis, and induction of oxidative stress [72,73,74,75]. In addition, metal ions such as zinc, iron, and copper have been shown to exacerbate the aggregation of Aβ-amyloid, α-synuclein, prion protein, or ataxin-3 triggering neurodegeneration [1,76,77,78]. Iron and copper are of significant interest, because they can catalyze production of toxic reactive oxygen species (ROS) through Fenton chemistry [79,80,81]. In this reaction, reduced iron and copper participate in the production of hydroxyl radicals that damage proteins, DNA, and lipids through oxidative modifications. Copper plays a central role in the well-known cuprizone animal model of MS in which the copper chelator induces its neurotoxic effect through oligodendrocytes death and subsequent demyelination [14]. Metals can also bind to sulfhydryl groups (-SH) and OH, Cl and NH_2_ in proteins, enzymes, coenzymes, and cell membranes inducing dysfunctions of the immune system [82]. It has also been proposed that copper levels may regulate the shift between anti-inflammatory and pro-inflammatory phenotypes in microglia via the regulation of nitric oxide levels and disruption of S-nitrosothiol signaling [83,84]. For instance, post-mortem analyses of amyloid plaque lesions in AD have reported an accumulation of copper, iron, and zinc compared to the normal brain [85,86]. A 339% increase in zinc, 466% increase in copper, and 177% increase in iron have been found in the plaques in comparison to the healthy subjects [87]. In these lesions, copper–amyloid complexes are expected to catalyze the production of ROS involved in neuronal death [88,89].

### 3.2. Xenobiotic and Essential Metals Interplay

Additionally, environmental xenobiotic metals such as mercury, lead, aluminum, arsenic, and cadmium are other candidate that may enter the organism and interfere with biological chemistry in the brain. These xenobiotic compounds have been detected in the normal brain with aluminum levels as high as 100 µg/g [90]. New applications and manufacturing processes increasingly expose humans to a number of metals to which they have not been exposed to in the past. Common sources of the metals are mining, burning of fossil fuels, tailings, industrial waste, agricultural runoff, paints, treated timber, aging water supply infrastructure, vehicle emission, lead-acid batteries, electronic components, fertilizers, and microplastics. They may be found in organic or inorganic chemical compounds, such as solvents and cytotoxic substances (e.g., pesticides, cigarette smoke, diesel exhaust particles) [91]. Moreover, environmental and occupational exposure to one metal is likely to be accompanied by exposure to other metals as well, and it is expected that interactions between different metals may occur in populations exposed to mixtures of metals with other chemicals. The main route of human exposure to these metals include food, drinking water, metal-containing drug ingestion, inhalation, and dermal contact. These metals generally have long biological half-life due to the lack of bodily recognition system. The levels of these toxic metals in the blood have been associated with polymorphisms in metal transporter genes, thus providing evidence that subsets of individuals are more susceptible to the toxic effects of some metals [92].

Xenobiotic metals interfere with the biological activity of proteins through diverse mechanisms. They may (1) displace essential metal ions and compete for their binding sites; (2) bind to free thiol (-SH) and selenol groups of proteins; (3) catalyze oxidation of amino acid side chains; (4) interfere with the folding of protein into 3D structure; (5) prevent their refolding [93]. These interferences affect metal transporters and other proteins and promote energy failure due to mitochondrial dysfunction, protein damage/aggregation, and metabolic alterations that challenge neuroglial function and trigger excitotoxic and inflammatory processes [94]. Oxidative stress caused by ROS is also a well-known mechanism of xenobiotic metal-induced damages [95,96]. Moreover, ROS leads to imbalance in homeostasis between antioxidant and pro-oxidant molecules and results in oxidative stress-related damage to cellular components such as proteins, DNA, and lipids [96].

For example, the most abundant form of mercury in the human body, known as organic methylmercury, enters the organism through pelagic species (fish and shellfish) consumption. This metal comes from inorganic mercury methylated by microorganisms from aquatic environment, and it is known to present high affinity for sulfhydryl (thiol) and selenohydryl (selenol) groups in proteins and to cross the blood–brain barrier by binding onto thiol groups of proteins; it can also bind to lone cysteine, mimicking the structure of methionine allowing for the uptake by amino acid transporters [97,98]. Several studies have established that this metal leads to depletion of intracellular antioxidants such as glutathione (GSH) and inhibition of several enzymes [99]. Aluminum is known to amplify catalytic pro-oxidant reactions involving essential metals [100,101,102]. This metal exposure causes the disruption of iron homeostasis leading to iron overload and oxidative stress [103,104]. This metal may also decrease ferritin synthesis and increase the expression of transferrin receptors, thereby increasing free iron levels in the cell resulting in an increase in oxidative damage via the Fenton reaction [105]. Aluminum has been shown to bind to ferritin and compete with iron for its binding to the transporter fransferrin and to induce oxidative damage, which can lead to neurodegeneration in the animal model [106,107]. Aluminum has also been shown to influence calcium homeostasis and calcium-dependent processes in the brain such as programmed cell death in astrocytes [108]. Animal studies have also shown that aluminum exposure affects permeability of the blood–brain barrier, cholinergic activity, signal transduction pathways, and impair neuronal glutamate nitric oxide-cyclic GMP pathway [109,110]. Aluminum exposure also decreases reduced gluthatione (GSH) levels and the activities of catalase, superoxide dismutase, glutathione peroxidase (GPx), and glutathione reductase and increases the levels of nitric oxide (NO) [111,112]. Thus, oxidative stress and subsequent mitochondrial dysfunction constitute the major vehicle underpinning aluminum-induced neurotoxicity (reviewed by Kumar et al. [113]). Aluminum can also activate microglia leading to secretion of TNF-alpha, IL-6, and cytokine-inducible nitric oxide synthase (iNOS or NOS-2) and the induction of proinflammatory cytokines [114]. Interestingly, high concentrations of this metal have been reported in the brain of patients with AD [115], and recently, a meta-analysis has shown that chronic exposure to it increases the risk of the disease [116]. Lead, another xenobiotic metal, has proved acute toxicity by interfering with the metabolism of essential metals, particularly that of calcium, iron, copper, and zinc [117,118]. Through its binding to thiol groups of both reduced glutathione (GSH) and oxidized glutathione (GSSH), this metal can also disturb the GSH/GSSH balance and renders cells more prone to oxidative damage [119]. Cadmium is another element with considerable toxicity for humans that may be taken up in rather high concentrations via edible parts of plants [120]. Cadmium accumulates with age in metallothioneins, which are involved in zinc and copper metabolism, as mentioned above [120,121,122]. Interestingly, mercury, lead, and cadmium metals have also been shown to exacerbate autoimmunity in the animal model. Similar to all xenobiotics, these metals may have a direct or indirect immunomodulatory effect on subpopulations, such as T helper type 1 (Th1), type 2 (Th2), and type 17 (Th17) cells and regulatory T (Treg) cells, thus disturbing the balance of otherwise finely tuned immune reactions [123,124,125,126,127,128].

## 4. Metals Imbalance in Multiple Sclerosis Patients

### 4.1. Metals in the Blood or Serum

The deregulation of metal homeostasis as a cause of MS is still a matter of debate [129,130,131,132]. Literature data concerning metal alterations in MS patients are partly contradictory and may depend on the biological specimens used for analysis (such as blood, serum, hair, urine, CSF, or post-mortem tissues from autopsy) making comparisons between studies of limited values. Several studies using atomic absorption spectrophotometry, the colorimetric method, or neutron activation analysis have reported altered metal levels in blood or serum from MS patients, such as decreased levels of zinc [132,133,134], increased levels of copper [135,136], increased levels of arsenic and lead [137,138], and increased levels of mercury [139] compared to healthy control groups.

Inductively coupled plasma mass spectrometry (ICP-MS) technology has also been used in some studies to analyze simultaneously several metals in MS patients. A chemical element profile has been reported by Alimonti et al. using ICP atomic emission spectrometry and ICP-MS in a series of sixty relapsing–remitting and secondary progressive patients [140]. In this study, the authors found a complex metal imbalance, notably low iron and zinc levels and high levels of calcium, zirconium, cadmium, tungsten, antimony, silicon, and nickel elements in serum of MS patients compared to healthy controls. They found that this imbalance was associated with an increase in serum oxidative status and with a decrease in serum anti-oxidant capacity [140]. In 2017, Janghorbani et al. reported in a series of 55 MS patients a lower concentration for calcium, iron, potassium, sulfur, sodium, phosphorus, and zinc as well as increased concentrations of copper and silicon in the blood of MS patients compared to healthy individuals [141]. Recently, a study designed by De Oliveira et al. investigating several metal elements by ICP-MS technology in MS patients has reported lower blood concentrations of beryllium, copper, chromium, cobalt, nickel, magnesium, and iron metals and higher concentrations of lead compared to healthy subjects [142]. Another recent study conducted by Siotto et al. has also reported in the peripheral blood of a series of 60 relapsing–remitting MS patients’ higher copper levels, decreased antioxidant capacity, and increased oxidative status compared to healthy controls [143]. Interestingly, these authors also found that iron levels were higher in the group of untreated MS patients compared to controls and patients receiving interferon-beta treatment suggesting a role of the therapy in oxidative stress related to iron metabolism [143]. Overall, these studies suggest that a set of different metals related to systemic oxidative status is linked to MS disease progression.

### 4.2. Iron in the Central Nervous System

Intratissue metal contents in the brain of MS patients have only been published using post mortem specimens. Alterations in iron content and distribution in these tissues are complex and well documented. These iron-related findings have originally been established on brain tissue sections using histochemical methods by Hametner et al. [19]. Studies have shown a loss of iron in most active and inactive lesions compared to the surrounding normal appearing white matter (NAWM) [19,144,145]. Iron levels have been shown to decrease in the NAWM of MS patients compared to controls with increasing disease duration, presumably due to the destruction of iron-loaded oligodendrocytes [19,144]. Compared to age-matched control subjects, increased amounts of iron have been observed in the basal ganglia and motor cortex of MS patients [146,147,148]. Active lesions may contain variable numbers of iron-loaded macrophages and a low content in astrocytes [19,144,149]. Reactive astrocytes organized in large astrogliotic areas in a subset of smoldering and inactive plaques accumulate iron as ferrihydrite in ferritin in the brain of patients [144]. In addition, at the edges of slowly expanding and some inactive lesions, iron accumulation has been described within microglia/macrophages showing a pro-inflammatory immunophenotype (CD86 and p22phox) [19,149,150]. These iron-related changes in the MS brain are likely to depend on the source of iron (heme or non-heme iron), the cell-type-specific expression patterns of iron-transport molecules, and their alterations in the demyelinating and neurodegenerative process [26]. It appears clear that microglia play a critical role in the maintenance of brain iron homeostasis; however, more studies are needed to elucidate the mechanisms and conditions in which iron cycling by microglia contributes to or is protective against neural disorders. Interestingly, a study by Magliozzi et al. regarding proteomic technology has correlated intrathecal deregulation of iron homeostasis pathway to cortical damage at early disease stage in MS patients [151]. Moreover, brain iron content evaluated by quantitative MRI in deep gray matter structures has been correlated with levels of disability in MS patients [152]. In addition, iron content in deep gray matter in progressive MS patients have been associated with the presence of genetic variants associated with iron regulation and metabolism [153]. Overall, these studies highlight the well-established imbalance of iron metal in the brain of MS patients possibly due to deregulated trafficking of this metal.

### 4.3. Other Metals (Magnesium, Copper, Manganese, Zinc, Aluminum) in the Central Nervous System

Alterations in other metal contents in brain of MS patients are far less documented. Yasui et al. reported lower levels of magnesium in MS patients compared to the control group [154]. Studies have reported increased levels of copper and decreased levels of manganese in the CSF of MS patients compared to healthy control groups [135,155]. Popescu et al. found decreased zinc levels in most white matter lesions of MS patients [61]. In a meta-analysis, Bredholt et al. also reported higher levels of zinc in CSF of MS patients compared to healthy controls [156]. It is interesting to note that zinc metal has been linked to the regulation of matrix metalloproteinase (MMP) activity by a mechanism that was named “cysteine switch” [157]. Disruption of interaction between zinc and cysteine residue in the metal binding-site containing catalytic domain is known to induce the full activation of these enzymes. Interestingly, increased expression of two zinc-regulating metallothionein isoforms I+II (MT-I+II) has been previously shown in astrocytes and activated macrophages/microglia [158] as well as overexpression of matrix metalloproteinases MMP-2, 7, and 9 in astrocytes, endothelial cells, lymphocytes, and macrophages, especially around blood vessels within MS lesions [159,160,161,162,163,164]. Otherwise, high contents and questionable tissue deposits of aluminum have been reported in both white and grey matter of brain parenchyma from MS patients [165] and recently significant higher contents of this metal have been evidenced in these tissues compared to the control group [166]. Thus, it appears that iron and probably zinc metals are linked to MS pathophysiology, although the description of their precise role in this process requires further studies.

A schematic description of some putative xenobiotic-induced metal imbalance involved in physiopathological process in CNS of MS patients is depicted in Figure 1.

## 5. Elemental Imaging for Understanding the Complexity of Metal Biochemistry?

The chemical elements essential for the normal brain metabolism and functioning such as calcium, copper, zinc, or iron are known to be interdependent. For example, due to their physicochemical nature, zinc and copper can compete for the binding sites of the same transporters and metal-binding proteins resulting in an antagonistic relationship, where low levels of zinc increase copper levels and vice versa [167]. As previously mentioned, there are elements such as aluminum, which can modify the transport and uptake into cells of essential metal ions such as iron [100,101]. Due to these interactions, the increase or the loss of one metal can lead to the establishment of a completely new metal profile affecting many others [168]. Most investigations on metallic elements in MS disease have evidenced accumulation or depletion of some elements (more evident for iron and zinc) in different biological samples or post-mortem tissue specimens. However, these studies are still controversial, because most of them have considered a small number of elements and are hampered by methodological limitations in terms of analytical quality, such as reduced number of samples and the different techniques applied.

Understanding the complexity of metallochemistry in the living brain is critical for designing appropriate therapeutic interventions. The inventory of metals and their species, including metalloproteins and metalloenzymes, in biological samples is termed as the metallome and its analysis as metallomics [169]. The sensitivity and specificity achievable with modern technologies allows the investigation of metal toxicity such as environmental exposure to xenobiotic metals [170,171,172,173]. Moreover, these methods enable the imaging of metal species and metal-containing compounds in biological samples. In situ visualization of metals in cells and tissues may play a major part in describing and in depth understanding of the metal chemistry. The characterization of metal distribution (imaging or mapping) in biological tissues has been of interest for a long time and technologies are now enabling such detection, each one achieving different spatial resolution, element selectivity, detection limit, and analytical depth [174,175,176,177]. Most of the current imaging techniques rely on methods that employ metal-selective probes/chemical sensors, light/lasers, electrons, X-ray, energetic particles, or mass spectrometric detection to measure characteristic radiation [37,176,177]. These techniques can be broadly divided into three groups: spectroscopy imaging (MRI, Fourier transform infrared imaging, Raman imaging, confocal laser scanning microscopy, super resolution microscopy, X-ray microscopy, synchrotron X-ray fluorescence microscopy, X-ray absorption spectroscopy, coherent diffractive imaging, positron emission tomography, laser-induced breakdown spectroscopy, mass spectrometry imaging (ICP-MS, secondary ion mass spectrometry, matrix-assisted laser desorption/ionization mass spectrometry imaging), and particle beam microscopy (electron microscopy, particle-induced X-ray emission). MRI is an established biomedical imaging technique of iron enabling the analysis of the anatomy and physiological processes of the brain. It provides high-resolution tomographic images with excellent tissue contrast; its spatial resolution is at the organ level but does not reach the cellular level. Some of these imaging techniques have previously been applied to study metals within the CNS in ex vivo brain tissue sections [67,178]. These technologies supported the studies of the normal brain and of disease-mediated changes in the storage and metabolism of metal, which may occur in specific intracellular compartments [179] or as widespread accumulation in multiple regions of the brain [180]. These imaging techniques have previously reported altered metal composition in brain tissue sections of patients with neurodegenerative diseases, such as AD, PD, and ALS [70,178,181,182] but not of MS patients, to our knowledge.

## 6. Conclusions and Perspectives

Metals are essential for human health and brain function. Alterations in their content or distribution are expected to exert neurotoxicity. Both alterations in the chemistry of essential metals and environmental exposure to xenobiotic metals can have silent chronic effects leading to neurodegeneration and neurological dysfunction. The restoration of metal homeostasis by using specific chelators has been considered in the past as a key pharmacological target in neurodegenerative diseases [23,183,184]. However, the complexity of the whole metal homeostasis in the brain has been largely unexplored, and only few clinical trials evaluating safety and therapeutic value of iron chelation therapies have been performed in the past in MS patients. Thus, further investigations are needed to describe broader metal biochemistry abnormalities in the brain and identify potential new therapeutic strategies in these patients. There is an increasing need for integrating safe and non-invasive analytical tools and imaging techniques to afford a complete picture of the metallome, including in vivo information on metal accumulation or depletion and distribution in the brain of MS patients for clinical purpose.

Our review should be considered as a prerequisite before undertaking in-depth complex transdisciplinary analytical studies on metals in the brain of MS patients. At a clinical level, such studies may provide in the future new perspectives for identification of metals classified as a risk factor or possible cause of the disease, ameliorating diagnosis, appreciating severity, and anticipating relapses as well as providing specific metal profiles indicative for a clinical response to innovative metal-targeted therapeutics. The targeting of one (or several) metal requires caution, by evaluating its specificity and its potential toxicity. The requirement for a metal chelator needs to be carefully considered, because high-affinity chelators may remove metals from “biological circulation” but may also trip metals from normal endogenous protein pathways. In contrast, moderate affinity chelators may have less impact on normal cellular processes. In addition, careful design may result in compounds that can effectively re-distribute metals from areas of excess to areas of deficiency, thereby reducing “toxic” events and also ensuring that physiological metal-dependent pathways remain functional. Development of such compounds referred to as “ionophores”, “modulators”, “chaperones”, or “metal protein attenuating compounds” must be encouraged.

## Figures and Tables

**Figure 1 ijms-21-09105-f001:**
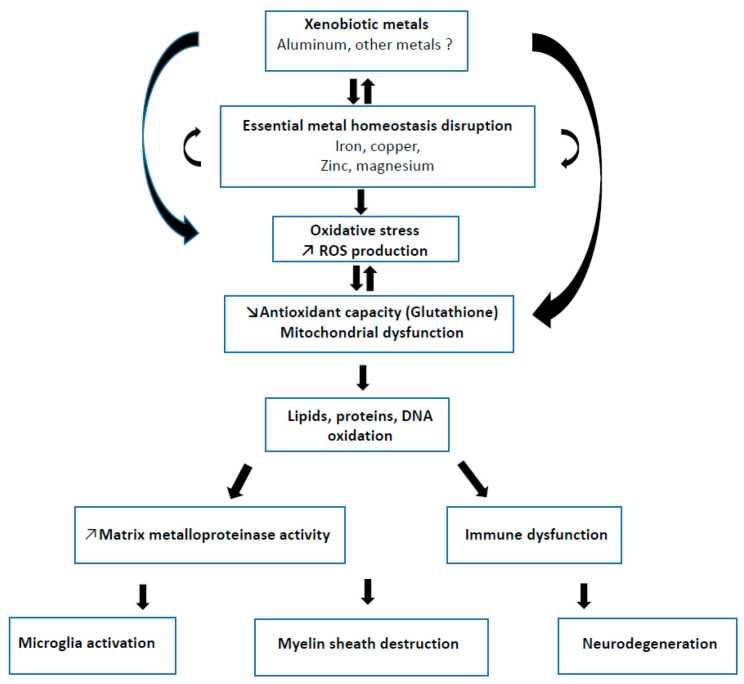
Schematic description of some putative xenobiotic-induced metal imbalance involved in physiopathological process in CNS of MS patients.

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
