# Peer review of "Metal Imbalance in Neurodegenerative Diseases with a Specific Concern to the Brain of Multiple Sclerosis Patients"

_ijms, 2020, doi:10.3390/ijms21239105_

Round 1

Reviewer 1 Report

Review on  "Metal Imbalance in brain of multiple sclerosis patients".

This review is an important analysis of the metals deregulation in Multiple sclerosis due to the importance of these topic in neurodegenerative disease. This field is a very important one that in my opinion deserve more attention and effort from scientific community. There are too many evidences of metals imbalance in CNS and of connection between this imbalance, oxidative stress and neuronal suffering and distress.

I appreciate the clear distinction between xenobiotic and essential metals as well as their important interplay, which is often a matter not well discussed. This is an important argumentation which could explain the interaction between environmental pollution in the modern society and the increase of neurodegenerative disease, often defined as multi-factorial diseases (referring to their complexity due to the interaction between genetic, metabolic and environmental causes).

I’m personally skeptic regarding  upcoming chelating therapies; at the moment we are far from understanding the real brain distribution of metals and its very sensitive and balanced (or unbalanced) regulation in CNS.  We are far from employ imaging techniques usefully to detect  brain metals distribution “in vivo”, having information on post-mortem brains that are quite  different respect to normal metabolism; moreover  chelation therapy seems not so specific for a single metals and more future efforts must be made from scientific community.

The review is well presented and references are in accordance with text, but a better organization of the paragraphs should be taken in consideration by Authors.

The major important modification is to create a dedicated paragraph on serum ICP metal analysis which is included in paragraph 4.2 but it seems out of “Metals in brain parencima” section.  (see pag 7, line 299-312). This part follows the part explaining techniques but it regards serum and not brain analysis. So, Authors should consider to rearrange better  this part.

Minor issues:

- dysregulation is not correct: the term is deregulation (abstract line 17)

- Please see reference   35 at pg 5 line 81, which seems not correspondent to the text.

-pg 3 line 100: when authors refer to spectroscopic methods , please add a specific paragraph indication (eg. Par xx. Pag xx) and remove  the “see below”.

-pg 3, line 103: the concentrations of calcium and magnesium are referred to grams of wet or dry brain? Authors should specify.

-pg 3 line 110 please correct Fe 3+ with Fe3+.

- pg 3 line 116: please correct the sentence: “they are the highest… “.. because it is not clear.

-pg 3, line 136 : please substitute NH2 with NH2

-pg 3 line 137: reference 79 seems not correct.

- pg4: remember to specify that we have only information on post-mortem brain

-pg4, line 173: methylmercury is an organic compound generated in the environment as a by-product of carbon-mercury bound, Authors should mentioned and explained that is not the inorganic metal form.

-pg 5 paragraph 4.1, probably it is more clear to divide paragraph on the basis of metals.  E.g. “Iron” , “other metals” or specify which metals , so that is too easy to read and to follow.

-pg 6, figure1: please ameliorate or remove the little arrows beside essential metal homeostasis box: what they indicate?

-pg 7, line 294 and 296: the sentences are not clear: authors should change as “These technologies ..”

-pg 7 lone 301: please add the number of patients analyzed from Alimonti et al. Please add a reference of Siotto et al (Siotto, 2019, Frontiers in Neuroscience) on Iron and oxidative stress imbalance in 60 Relapsing remittinf MS patients, in which authors found that iron was higher in patients not treated with interferon-beta and that oxidative stress was higher in all MS, together with lower antioxidant capacity.

- conclusion: it should be better discussed the future perspective and the current difficulty in this important field of research.

Reviewer 2 Report

This is a review of the metal imbalance in the brain of Multiple Sclerosis
patients. The article is a collection of a lot of information without any well-defined outline. It is very difficult to follow and very disorganized. It is not enough to separate the information by some subtitles. The article is not focused on any specific aim. The authors need to have a specific aim and outline for the review article. 

The introduction should be more focused on the title of the study to justify why reviewing this information are important. The only figure in the article is not informative. A table might be helpful if include all potential metals that affected MS with the suggested mechanisms.

At the end of the study, the preventive, clinical, or therapeutic importance of the metals in multiple sclerosis is still unclear.

Reviewer 3 Report

This work reviews an interesting topic in MS pathogenesis.  The role of metal imbalances in MS. 

Although this work is titled "Metal imbalances in the brain of multiple sclerosis patients", 103 lines are dedicated to reviewing the data on the role of metals in the pathogenesis of MS and  184 lines are introductory or discussing the role of metals in other CNS diseases. 

Review of available data on the putative role of metals in the pathogenesis of MS is by no means exhaustive. One of the most commonly used models of MS , Cuprizone-induced demyelination is induced by Copper chelation thought to primarily affect oligodendrocytes of the corpus callosum (Benardais et al, 2013, NeurotoxRes)  

Important work regarding iron in MS has not been presented e.g. (but not limited to):

  • 3T MRI techniques may differentiate MS from neuromyelitis optica lesions (another demyelinating disease)  based on the paramagnetic rim of demyelinated lesions containing Fe-filled microglia/macrophages. Thus, the iron- pattern of demyelinated lesions is rather specific for MS   Jang et al, 2020, JClin Neurol.
  • Proteomic analysis of CSF from MS cases with high cortical demyelination lesions loads revealed changes in proteins related with iron homeostasis Magliozzi et al, 2019, AnnClinTranslNeurol
  • iron content of deep gray matter structures in MS brains correlates with levels of disability Zivadinov et al, 2018, Radiol
  • Iron content in basal ganglia nuclei in MS may be associated with the presence of particular polymorphisms of proteins associated with iron homeostasis  Hegemeier et al, 2017, NeuroimageClin

Data is primarily "listed" and presented superficially and no clear hypothesis is formulated. Which metal do you think there is more convincing evidence it may have a role in the pathogenesis of MS? Does it affect the likelihood of getting MS, the relapse rate?, the severity of relapses?, could metal imbalance or iron accumulation be responsible for disability progression in the progressive forms of MS? A section would be useful to describe how the authors think that metal dyshomeostasis may contribute to specific pathological and clinical features of MS e.g. chronic demyelination?, chronic axonal damage?, chronic activation of microglia?, cortical lesion formation? progression of disability?     

In line 106, it is mentioned that iron accumulates with ageing.  Ageing is a key risk factor for the conversion of RR-MS to SP-MS and PP-MS typically manifests in older ages. Could iron accumulation have something to do with this? It would be useful to discuss. 

Minor points

line 2: Metal Imbalances in the brain of multiple sclerosis patients

line 13: causing or manifesting with or associated with more appropriate than provoking............ 

line 14: in the CNS associated with.............

line 25: in which an inflammatory process results.....

Round 2

Reviewer 2 Report

The papaer has been improved.

Reviewer 3 Report

The manuscript is now greatly improved with the changes made to it.